# The Dietary Inflammatory Index and Early COPD: Results from the National Health and Nutrition Examination Survey

**DOI:** 10.3390/nu14142841

**Published:** 2022-07-11

**Authors:** Chen Chen, Ting Yang, Chen Wang

**Affiliations:** 1Beijing University of Chinese Medicine, Beijing 100029, China; chenchen2020a@163.com; 2Department of Pulmonary and Critical Care Medicine, Center of Respiratory Medicine, China-Japan Friendship Hospital, Beijing 100029, China

**Keywords:** early COPD, dietary inflammatory index, lung function, NHANES

## Abstract

We examined 3962 people aged 20 to 49 years who had information on spirometry testing and underwent a 24 h dietary recall interview from the 2007–2012 National Health and Nutrition Examination Survey (NHANES) and used multivariable logistic regression to evaluate associations between Dietary Inflammatory Index (DII, a pro-inflammatory diet) and early COPD and lung function. The overall prevalence of early COPD was 5.05%. Higher DII was associated with increased odds of early COPD (quartile 4 vs. 1, the OR = 1.657, 95% CI = 1.100–2.496, *p* = 0.0156). In a full-adjusted model, each unit of increase in DII score was associated with a 90.3% increase in the risk of early COPD. Higher DII is significantly associated with lower FEV_1_ and FVC among individuals with early COPD, each unit increment in the DII was significantly associated with 0.43 L–0.58 L decrements in FEV_1_ (β = –0.43, 95% CI = −0.74, −0.12) and FVC (β = −0.58, 95% CI = −1.01, −0.16). These findings demonstrate that higher consumption of a pro-inflammatory diet may contribute to an increased risk of early COPD and lower lung function, and further support dietary interventions as part of a healthy lifestyle in order to preserve lung function and prevent or improve COPD.

## 1. Introduction

Chronic obstructive pulmonary disease (COPD), the third leading cause of death worldwide [1], is considered a disease of the elderly and little attention has been paid to the initial stages of the natural history of COPD. Current evidence shows that the patient’s lungs have already been irreversibly damaged before the occurrence of severe respiratory symptoms. As the disease progresses, the symptoms gradually worsen, and the effect of later intervention is limited [2,3]. Therefore, early diagnosis and treatment are the keys to the prevention and control of COPD, and early intervention can effectively slow down the decline rate of lung function and improve the prognosis. In addition, chronic inflammation plays a central role in the development of COPD.

A great deal of evidence shows that many foods, dietary factors, and non-nutritive components can regulate inflammatory states, both acutely and chronically [4,5]. Therefore, diet, as a modifiable lifestyle factor, may play an important role in anti-inflammation, improving lung function, and reducing the risk of COPD incidence and progression [6,7,8,9,10]. From this perspective, a method to characterize and measure the inflammatory potential of an individual’s diet could aid in the development of tailored and precise dietary interventions and health maintenance strategies. Among others, the Dietary Inflammation Index (DII^®^) [11], a literature-derived dietary tool, consisting of 45 items including energy, nutrients, and foods/spices, provides a comprehensive way to explore the relationship between the inflammatory potential of diet and different health-related outcome methods. A higher DII score reflects a pro-inflammatory diet.

Here, we used data representing the U.S. population to assess the impact of dietary inflammation on the risk of developing early COPD. We hypothesized that a pro-inflammatory diet is associated with an increased risk of early COPD, and worsened lung function.

## 2. Materials and Methods

### 2.1. Subject Population

National Health and Nutrition Examination Survey (NHANES) is an ongoing cross-sectional survey of the U.S. non-institutionalized civilian population, selected using a stratified multistage probability sampling design to derive a representative sample of the U.S. population. By design, persons 80 years and older and ethnic minorities were oversampled to increase the statistical power for data analysis. Individuals (age 20–49 years) who participated in the 2007–2008, 2009–2010, and 2011–2012 NHANES cycles were included in this analysis. NHANES protocols were approved by the Institutional Review Boards of the NCHS and the CDC, and informed consent was obtained from all participants. Details on NHANES procedures, methods and IRB approval are available (https://www.cdc.gov/nchs/nhanes/index.htm, accessed on 9 June 2022).

### 2.2. The Dietary Inflammatory Index (DII^®^)

The development and validation of the DII have previously been reported [11,12]. Briefly, each food and component was assigned a score by specific markers (IL-1β, IL-4, IL-6, IL-10, tumor necrosis factor-α (TNF-α) and C-reactive protein (CRP)) and multiplied by the food the individual actually consumed parameters, resulting in an overall score that summarizes individuals’ diets from maximally anti-inflammatory to maximally pro-inflammatory. Dietary data collected through 24 h dietary recall interviews or food records were used to calculate DII scores. In the present study, a total of 26 food parameters were available in NHANES and were used for the calculation of DII, including alcohol, energy, carbohydrates, proteins, fats, fiber, cholesterol, saturated fatty acid, mono-unsaturated fatty acid, poly-unsaturated fatty acid, niacin, vitamins (A, B1, B2, B6, B12, C, D and E), iron, magnesium, zinc, selenium, folic acid, beta carotene and caffeine. Figure 1 shows the sequence of steps to create the dietary inflammatory index (DII) for the NHANES 2007–2012.

### 2.3. Definition of Early COPD and Exposure History

Spirometry data were collected on participants aged 20 to 49 years who met strict inclusion criteria as detailed per NHANES Survey. The best forced expiratory volume in 1 s (FEV_1_) and forced vital capacity (FVC) were selected for analysis. The lower limit of normal (LLN) is defined as the bottom 5th percentile of the predicted value. Calculations of LLN and % predicted for FEV_1_, FVC and FEV_1_/FVC were performed based on sex, race, height and age calculations according to Hankinson et al. [13]. Early COPD was defined as a FEV_1_/FVC < LLN in participants aged less than 50 years old [14]. Individuals with FEV_1_/FVC < 0.70 were excluded (Figure 1). 

Information on smoking and occupation exposure were obtained from the questionnaire. Smoking status was defined as never, former, or current smoking. “Have you smoked at least 100 cigarettes in your entire life?” Participants who answered “No” were classified as “never smokers”. Those who answered “Yes” were identified as smokers, and based on their answer to the question, “Do you smoke cigarettes now?” they were classified as “current smokers” (“Yes”) or “former smokers” (“No”). Cigarette consumption was calculated in pack-year based on information on age at smoking initiation and cessation (or for current smokers until age at examination): “How old when you first started to smoke cigarettes fairly regularly?”, “How long has it been since you quit smoking cigarettes?”, “How many cigarettes did you smoke per day?”, “How many cigarettes did you usually smoke per day when quit?”. A pack-year was defined as 20 cigarettes smoked daily for a year. Individuals with and without early COPD were subsequently stratified according to smoking exposure: smokers with ≥10 pack-years, smokers with <10 pack-years, and never-smokers.

The NHANES also asked the questions, “Does anyone smoke in your home?” and “At this job or business, how many hours per day can you smell the smoke from other people’s cigarettes, cigars, and/or pipes?”. We defined passive smoking as inhalation of smoke by non-smokers who lived or worked with smokers. Occupation exposure was defined by a positive answer to those questions: “In any job, have you ever been exposed to dust [from (rock, sand, concrete, coal, asbestos, silica or soil) or (baking flours, grains, wood, cotton, plants or animals)], exhaust fumes (from trucks, buses, heavy machinery or diesel engines) or any other gases, vapors or fumes (from paints, cleaning products, glues, solvents, and acids; or welding/soldering fumes)?”.

### 2.4. Respiratory Diseases, Chronic Respiratory Symptoms and Other Characteristics

Emphysema, chronic bronchitis or asthma diagnosis was defined by a positive response to the question, “Has a doctor or other health professional ever told you that you had emphysema, chronic bronchitis or asthma?”. We defined history of childhood emphysema, bronchitis or asthma before age 14 years by the question, “How old were you when you were first told had emphysema, bronchitis or asthma?”. NHANES participants ≥ 40 years old were also asked about respiratory symptoms as part of the NHANES Household Questionnaire Interview. The questions included cough ≥ 3 mo during the year “Do you cough on most days for ≥3 consecutive months during the year?”; phlegm ≥ 3 mo during the year, “Do you bring up phlegm on most days for 3 consecutive months or more during the year?”; wheezing in the past year, (In the past 12 mo, have you had wheezing or whistling in your chest?”; and shortness of breath, “Have you had shortness of breath either when hurrying on the level or walking up a slight hill?”.

Data on age, sex, race/ethnicity, BMI, educational level, family income (poverty income ratio), insurance coverage and sedentary activity minutes per day were collected using questionnaires. Poverty income ratio (PIR) was estimated using guidelines and adjustments for family size, year and state. Participants were dichotomized by PIR into levels below and above 1.85, the cutoff for Supplemental Nutrition Assistance Program (SNAP) eligibility. 

### 2.5. Statistical Analysis

All data were processed by Empower (R) (www.empowerstats.com, accessed on 9 June 2022; X&Y Solutions, Inc., Boston, MA, USA) and statistical package R (http://www.Rproject.org, accessed on 9 June 2022, The R Foundation) with a significance threshold of 2-sided *p* < 0.05. Continuous variables were presented as mean ± standard deviation (SD). Unpaired *t* test was used when data conformed to normal distribution and homogeneity of variance. Wilcoxon testing was used when data did not meet the condition. Categorical variables were presented as frequencies (percentages) and were compared with the Chi-square test.

To explore the association between DII and early COPD, we applied logistic regression models. Model 1, no covariate was adjusted; model 2, variables that were statistically significant in the univariate analysis were adjusted: age; sex; PIR; health insurance coverage; sedentary activity; smoking exposure (pack-years); mineral dusts; exhaust fumes; wheezing; asthma. There is significant collinearity between “Smoking exposure (pack-years)” and “Smokers living in the home”, “History of asthma” and “History of asthma during childhood”. Thus, “Smokers living in the home” and “History of asthma during childhood” were excluded. To draw a more confident conclusion from the statistical analysis, we performed a sensitivity analysis as shown below. First, we converted DII into a categorical variable (quartile) to determine if there is a non-linear relationship. Then linear trends were performed to verify the results of DII as a continuous variable. Covariates adjusted in model 2 were selected for subgroup analysis.

To explore the association between DII and lung function, we applied linear regression model, which adjusted for age; sex; PIR; health insurance coverage; sedentary activity; smoking exposure (pack-years); mineral dusts; exhaust fumes; wheezing; asthma. We also removed “Smokers living in the home” and “History of asthma during childhood”. And according to the presence or absence of early COPD, a subgroup analysis of the participants was carried out.

## 3. Results

### 3.1. Characteristics of Individuals

The demographics and general characteristics of the study population are described in Table 1. A total of 3962 people (2041 men and 1921 women) aged 20 to 49 years were included in the final analysis from NHANES 2007–2012 according to the information on spirometry testing and 24 h dietary recall interviews. The overall prevalence of early COPD was 5.05%. The prevalence decreased with age and was 50.50% among individuals aged 20–29 years and 16.50% among those aged 40 years or older in the early COPD population (Figure 2A). In addition, we found that the prevalence of early COPD was 6.13% in smokers with ≥10 pack-years, 5.73% in smokers with <10 pack-years, and 4.50% in never-smokers. Compared to individuals without early COPD, individuals with early COPD were more likely to be female (58.00% vs. 47.98%, *p* = 0.0057), to have a lower household income (56.91% vs. 46.49%, *p* = 0.0053), to lack health insurance coverage (41.00% vs. 34.47%, *p* = 0.0588), to have more smoking exposures, more specifically, the current smoking (35.68% vs. 24.69%, *p* = 0.0016) and smokers living in the home (48.36% vs. 31.29%, *p* = 0.0005) and to have lower lung function (Figure 2B). In addition, respiratory diseases during childhood (32.00% vs. 25.60%, *p* = 0.0441), asthma history (26.50% vs. 12.56%, *p* < 0.0001) and wheezing (20.50% vs. 9.87%, *p* < 0.0001) were also associated with increased prevalence of early COPD in the general population. The DII was significantly higher in individuals with early COPD than in those without early COPD (3.82 ± 0.26 vs. 3.76 ± 0.25, *p* < 0.0001).

### 3.2. Association between DII and Early COPD 

The association between DII and early COPD is shown in Table 2. Model 1, an unadjusted model, indicated that early COPD positively correlated with DII scores, each unit of increased DII score was associated with 1.361 times increased risk of early COPD. In a multivariable analysis of quartiles of DII, the fourth quartile of the DII had significantly higher odds of early COPD than those in the first quartile (quartile 4 vs. 1, the OR = 1.657, 95% CI = 1.100–2.496, *p* = 0.0156). In Model 2, the association between exposure variables and outcomes was still stable after adjusting for age, sex, PIR, health insurance coverage, sedentary activity, smoking exposure (pack-years), mineral dusts, exhaust fumes, wheezing and asthma history. One-point increment in the DII was significantly associated with 0.9 times (90.3%) increased odds of early COPD. Furthermore, our results indicated that the association between the higher DII and early COPD was more pronounced among those who were smokers with <10 pack-years, having no mineral dust exposure but wheezing, having respiratory diseases during childhood, as well as in those who had a lower household income (Table 3).

GAM model was used to evaluate the linear relationship between DII and early COPD. The result was positive, which proves that exposure variables and outcomes are related (Figure 3).

### 3.3. Multivariable Analysis of the DII and Lung Function

The multivariable analysis of the DII and lung function parameters among participants indicated that lung function negatively correlated with DII scores (Table 4). The DII was significantly associated with lower FEV_1_ (β = −0.39, 95% CI = −0.47, −0.31), FEV_1_% (β = −4.15, 95% CI = −5.85, −2.45), FVC (β = −0.56, 95% CI = −0.66, −0.45), FVC% (β = −9.05, 95% CI = −11.41, −6.7) and FEV_1_/FVC (β = 0.01, 95% CI = 0.01, 0.02) among individuals without early COPD. In early COPD, each unit increment in the DII was significantly associated with 0.43 L–0.58 L decrements in FEV_1_ (β = −0.43, 95% CI = −0.74, −0.12) and FVC (β = −0.58, 95% CI = −1.01, −0.16). The DII was not correlated with FEV_1_%, FVC% or FEV_1_/FVC in early COPD.

## 4. Discussion

To reduce COPD’s long-term societal impact, changing the goal of interventions from the sole intent of reducing symptoms, exacerbations and complications in advanced disease to halting progression in early disease is crucial. In recent years, researchers have proposed the concept of “early COPD”. Cigarette smoking is the most important environmental risk factor for COPD, and has been included in some definitions of early COPD [3]. Smoking is not the only cause of COPD, and we should emphasize the public health implication of interventions for unhealthy lifestyle habits.

To our knowledge, this is the first report of an association between a high DII score (indicating a pro-inflammatory diet) and early COPD. In this cross-sectional study of 3962 adults, a positive significant relation between DII and early COPD and a negative association with lung function were observed, indicating that a higher intake of a pro-inflammatory diet may contribute to an increased risk of early COPD and lower lung function. The association between the exposure variable and outcome variables was still stable after adjustment for covariates. Subgroup analysis stratified by the sex, exposure history, medical history and other variables showed that this correlation could be applied to the population with different environmental pollutant exposure conditions, wheezing status, childhood history of respiratory disease and household income status.

The dietary quality and nutritional status of COPD patients as well as the oxidative-inflammatory pathogenic basis of COPD provided evidence for validating the respiration effects of anti-inflammatory and antioxidant dietary components. Among the DII food parameters, nutrients mainly derived from fruits and vegetables (such as vitamins A, C, D, E, β-carotene, polyphenols and fiber) exhibit anti-inflammatory and antioxidant effects. Vitamin A is involved in the proliferation and maintenance of respiratory epithelial cells, is a major factor in regulating lung differentiation and maturation, and can also regulate local immunity and reduce inflammatory responses [15]. Vitamin E is a fat-soluble antioxidant that protects polyunsaturated fatty acids in cell membranes from being oxidized, regulates the production of reactive oxygen species and reactive nitrogen species, and regulates signal transduction [16]. Vitamin D deficiency is highly prevalent in COPD patients [17]. Current researches show that vitamin D plays a prominent role in the occurrence and development of COPD [18,19,20,21] by regulating genes associated with inflammation [22], differentiation and functions of immune cells [23], antimicrobial peptide production [24] and lung remodeling [23]. Numerous studies have shown that a dietary intake of fruits and vegetables was associated with decreased risk of COPD [7,8,9,10] and improved lung function [25,26,27,28]. In two large population-based prospective cohort of Swedish women [9] and men [10], high long-term consumption of fruits (in both men and women) and vegetables (only in men) were inversely and independently associated with the incidence of COPD (37% lower risk in women, 95% CI: 25–48%, *p* < 0.0001; 35% lower risk in men, 95% CI: 24–45%, *p* < 0.0001), particularly in pronounced in smokers. A higher Alternative Healthy Eating Index (AHEI-2010) score (higher score = better diet quality) was independently associated with better FEV_1_ and FVC and with lower odds of spirometric restriction [28]. A study of adults over 40 years old in NHANES, COPD individuals had higher DII scores than non-COPD individuals (0.429 ± 1.809 vs. −0.191 ± 1.791, *p* < 0.001) [29]. However, no studies have explored the association between DII and early COPD. We show that the DII score was positively correlated with the occurrence of early COPD.

We all know that changes in the diversity and abundance of microbial are closely related to the development and progression of COPD [30]. Diet is an important regulatory factor affecting the microbial composition and related metabolites. In addition to the direct modulation of lung microbiota by diet, more and more evidence has recently begun to reveal the impact of many foods, dietary factors, non-nutritive components and gut microbiota on lung function and COPD [31,32]. Among the food parameters included in the DII, saturated fatty acid (SFAs), trans-fat, cholesterol, and Omega-6 polyunsaturated fatty acids (n-6 PUFAs) are the main pro-inflammatory contributors, while Omega-3 polyunsaturated fatty acids (n-3 PUFAs) has anti-inflammatory properties [33,34]. The evidence suggests that saturated fatty acids induce inflammation in part by mimicking the actions of LPS [33]. LPS-mediated signaling through TLR4 leads to the activation of NF-κB, a transcription factor, which subsequently turns on the expression of numerous pro-inflammatory cytokines, such as TNF-α, IL-1, IL-6, and IL-8 [35]. A high-fat diet exacerbated the lung inflammation and microbial profile changes induced by diesel exhaust particles in mice [36]. Much has been written about the potential health benefits associated with increasing our intake of n-3 PUFAs [37,38,39]. However, n-3 PUFAs are so much less anti-inflammatory in humans than in mice [40], and the available literature suggests a weak benefit of n-3 PUFAs in COPD patients related to clinical outcomes [41,42]. The underlying mechanism through which a high-fiber diet plays a beneficial role in lung health with a “diet–gut microbiome–mucous barrier and immune disorders” link has been researched extensively [43]. SCFAs are considered one of the key microbial metabolites in this link [44]. A murine model demonstrated that a high-fiber diet increased the local SCFA concentration and improved inflammation, alveolar destruction and cellular apoptosis [45]. The dietary intake of decreased fiber has been associated with a newly developed airflow limitation among adults with normal spirometry [46]. In addition, SCFAs synergistically promote the synthesis of the host defense peptide (HDP) with vitamin D, a component of the innate immune system with immunomodulatory and antimicrobial activities [47,48].

Our results further complement current literature supporting the link between pro-inflammatory dietary patterns and COPD. The Western dietary pattern was characterized by higher consumption of pro-inflammatory foods such as red meat, cured (bacon, hot dogs and processed meats), foods with a high glycemic index (refined carbohydrates, sweetened beverages and sweets), fried food, margarine, etc. Long-term processed red meat consumption increases the risk of COPD [49,50]. Dietary advanced glycation end products (AGEs) are highly oxidant and pro-inflammatory compounds, with the highest levels present in cooked (grilling, frying and roasting) meats [51]. AGEs are ligands for the AGE receptor (RAGE), as one of the pattern recognition receptors (PRR) expressed by airway epithelial cells, with the highest expression in the lung tissues of COPD patients, and an important driving force of inflammation and pathophysiology [52]. A study of a national pediatric population in NHANES demonstrated an association between dietary AGE intake and wheezing symptoms (adjusted OR 1.18; 95% CI 1.02 to 1.36) [53]. On the contrary, the Mediterranean dietary pattern, characterized by a predominantly plant-based diet, with olive oil being the main type of added fat, may have important anti-inflammatory effects [6]. The latest prospective study has revealed that the “balanced dietary pattern” characterized by a high intake of fresh fruit and protein-rich foods (soybeans, meat, poultry, fish or seafood, eggs, and dairy products) is relatively associated with a lower risk of COPD in Chinese adults [54]. Overall, the evidence concordantly indicated that the dietary intake pattern is an important factor in the pathogenesis and prevention of COPD, involving the relationship between diet and symptoms, lung function, the risk of incidence and progression of COPD, and complications (depression) [55,56,57,58]. Consistent with the pro-inflammatory effects of a Westernized diet on the airways, we show that a higher DII is significantly associated with lower FEV_1_ and FVC among individuals with early COPD in NHANES.

We chose 10 pack-years for subgroup analysis [59], showing that higher DII may contribute to an increased risk of early COPD among smokers, particularly pronounced among those who were smokers with <10 pack-years. Since occupational dust exposure has a large effect on COPD, the association of DII with early COPD was not significant in subgroups of mineral dusts. Different types of occupational dust exposure may have different effects. Quality of dietary intake has been associated with socioeconomic disparities [60]. Low-income families, often rely on high energy-dense, and unhealthy foods (high in sugars and fats) and a low consumption of fruits and vegetables [60]. We have already discussed that a low-quality diet is associated with a higher risk of COPD and worse lung function. The association between DII and early COPD was found to be more pronounced among those who had a higher household income. Greater focus has been placed recently on the potential for early life factors to influence the development of COPD. Current research shows that childhood asthma, bronchitis, pneumonia, allergic rhinitis and eczema increased the risk of developing COPD by middle age [61]. Our results indicated that the higher DII increased the risk of early COPD in those who have a history of emphysema, bronchitis or asthma during childhood.

Strengths of the present study include a large sample, clinical characteristics of early COPD, and furthermore, our study is the first to examine the association of DII and early COPD and lung function in a nationally representative sample of US adults, providing a foundation for subsequent longitudinal studies to assess DII as a modifiable dietary risk factor in the development of inflammatory airways disease.

There are several limitations. First, causality cannot be implied, and we cannot determine a temporal relationship between the DII and early COPD or lung function, including early COPD and clinical COPD morbidity, COPD mortality and lung function trajectory due to the cross-sectional design. Second, post-bronchodilator spirometry was not used to identify early COPD at the baseline examination. Thus, some individuals, including asthmatics, may have a reversible form of airflow limitation. In addition, although the concept of “early COPD” was proposed for the first time in the 2022 Global Initiative for Chronic Obstructive Lung Disease (GOLD) Report, which is only appropriate for discussing “biological early” [62], the clear diagnostic criteria of early COPD are still unclear. Fourth, we did not analyze the data of WBCs, RAGE and other pro-inflammatory cytokines, so the correlation between DII, WBCs, RGAE, other pro-inflammatory cytokines, and early COPD cannot be clarified. Previous studies have shown that a pro-inflammatory diet is associated with increased white blood cell counts [63], and CRP is a biomarker for assessing COPD exacerbations [64]. Fourth, dietary intake was based on one 24 h dietary recall, which has a certain recall bias. Fifth, our DII was not compared with other dietary indexes such as the energy-adjusted DII (E-DII) [65], AHEI-2010 [28] and food insecurity (FI) [66].

## 5. Conclusions

In summary, our study suggests that a pro-inflammatory diet is associated with a higher risk of early COPD, and lower lung function. Given COPD is a major worldwide public health issue, identification of modifiable risk factors for the prevention and treatment of COPD from the perspective of population medicine is highly in demand. Importantly, given the early origin of COPD and the profound impact of diet on lung function and later respiratory health, dietary interventions offer opportunities for early strategies of primary prevention and/or targeted early therapeutic approaches. Moreover, while developing good eating habits, it is also necessary to combine with healthy lifestyles such as smoking cessation and maintaining a healthy weight to achieve better preventive effects.

## Figures and Tables

**Figure 1 nutrients-14-02841-f001:**
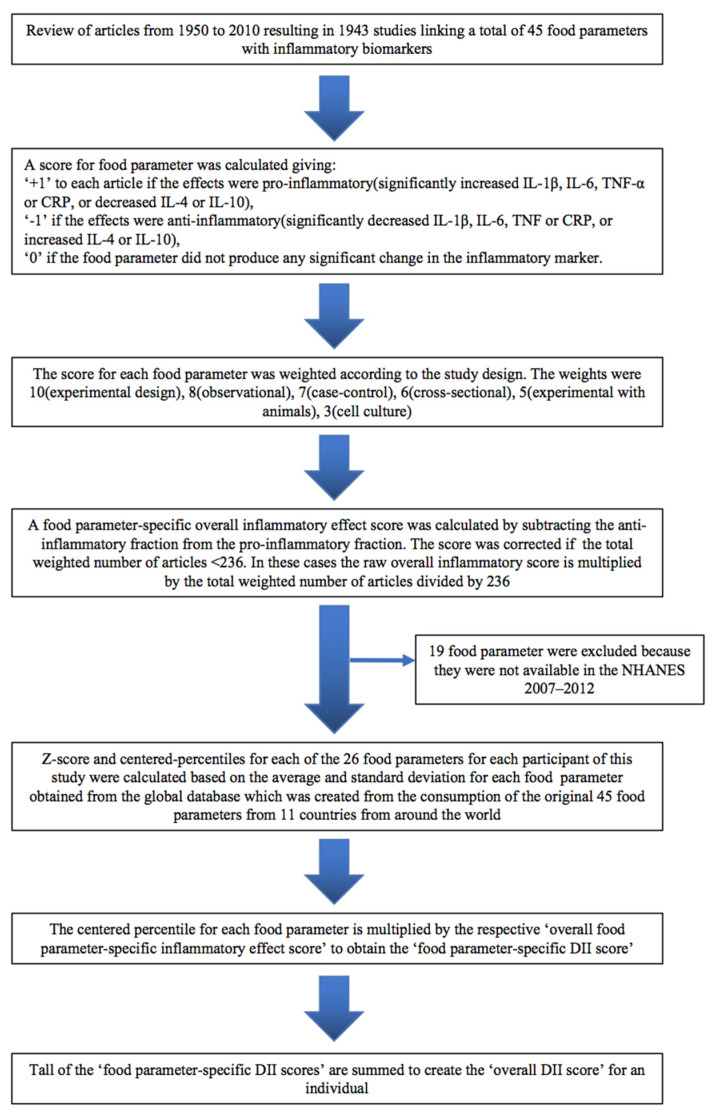
Flow diagram of study participants.

**Figure 2 nutrients-14-02841-f002:**
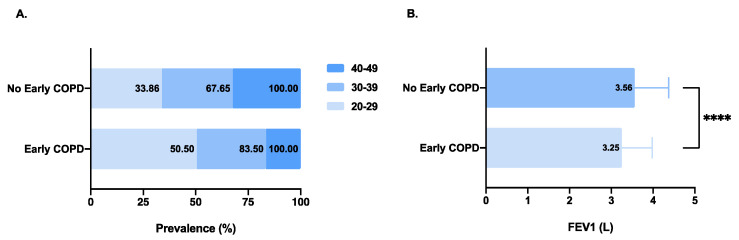
Differences in age and FEV_1_ between individuals with and without Early COPD. Notes: (**A**) Proportion of individuals aged 20–29, 30–39 and 40–49 at baseline examination. (**B**) Differences in FEV_1_ between individuals with and without Early COPD. *p*-Value: **** *p* < 0.0001.

**Figure 3 nutrients-14-02841-f003:**
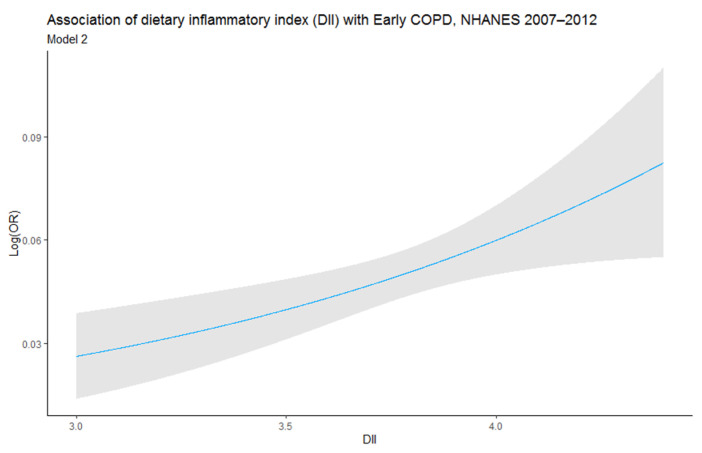
Relationship between dietary inflammatory index (DII) and Early COPD. Risk of Early COPD (blue) with 95% CIs (grey) determined using the generalized additive model. The model is adjusted for age, sex, PIR, health insurance coverage, sedentary activity, smoking exposure (pack-years), mineral dusts, exhaust fumes, wheezing and asthma.

**Table 1 nutrients-14-02841-t001:** Characteristics of individuals with and without Early COPD, NHANES 2007–2012.

Characteristics	Early COPD (*n* = 200)	No Early COPD (*n* = 3762)	*p*-Value
Demographics			
Age ****			<0.0001
20–29	101 (50.50)	1274 (33.86)	
30–39	66 (33.00)	1271 (33.79)	
40–49	33 (16.50)	1217 (32.35)	
Sex **			0.0057
Females	116 (58.00)	1805 (47.98)	
Males	84 (42.00)	1957 (52.02)	
Race/ethnicity			0.4984
Non-Hispanic white	96 (48.00)	1944 (51.67)	
Non-Hispanic black	53 (26.50)	980 (26.05)	
Mexican American	51 (25.50)	838 (22.28)	
Body-mass index (kg/m²)			0.7381
<18.5 (underweight)	2 (1.00)	56 (1.49)	
18.5–24.9 (normal weight)	67 (33.50)	1135 (30.18)	
25–29.9 (overweight)	63 (31.50)	1213 (32.25)	
≥30 (obese)	68 (34.00)	1357 (36.08)	
Missing	0	1	
Educational level			0.7413
Primary school and less	15 (7.54)	245 (6.52)	
Middle and high school	78 (39.20)	1421 (37.79)	
College and higher	106 (53.27)	2094 (55.69)	
Missing	1	2	
PIR **			0.0053
<1.85	107 (56.91)	1624 (46.49)	
≥1.85	81 (43.09)	1869 (53.51)	
Missing	12	269	
Health insurance coverage			0.0588
No	82 (41.00)	1296 (34.47)	
Yes	118 (59.00)	2464 (65.53)	
Missing	0	2	
Sedentary activity			0.1867
<3 h	53 (26.50)	802 (21.35)	
3–6 h	72 (36.00)	1367 (36.40)	
≥6 h	75 (37.50)	1587 (42.25)	
Missing	0	6	
Exposure history			
Smoking status **			0.0016
Never smoker	108 (60.34)	2290 (71.16)	
Former smoker	20 (10.05)	541 (14.39)	
Current smoker	71 (35.68)	928 (24.69)	
Missing	1	3	
Smoking exposure (pack-years)			0.1617
0	108 (55.96)	2290 (62.69)	
<10	54 (27.98)	888 (24.31)	
≥10	31 (16.06)	475 (13.00)	
Missing	7	109	
Passive smoking			0.4309
No	85 (85.86)	1628 (82.81)	
Yes	14 (14.14)	338 (17.19)	
Missing	101	1796	
Smokers living in the home ***			0.0005
0	63 (51.64)	1421 (68.71)	
1–2	51 (41.80)	561 (27.13)	
≥3	8 (6.56)	86 (4.16)	
Missing	78	1694	
Mineral dusts **			0.0018
No	151 (76.65)	2393 (65.89)	
Yes	46 (23.35)	1239 (34.11)	
Missing	3	130	
Organic dusts			0.8919
No	150 (76.14)	2750 (75.72)	
Yes	47 (23.86)	882 (24.28)	
Missing	3	130	
Fumes from machinery or engines **			0.0082
No	161 (81.73)	2662 (73.21)	
Yes	36 (18.27)	974 (26.79)	
Missing	3	126	
Any other gases, vapors or fumes			0.4033
No	139 (70.56)	2461 (67.70)	
Yes	58 (29.44)	1174 (32.30)	
Missing	3	127	
Medical history			
History of emphysema, bronchitis or asthma during childhood *			0.0441
No	136 (68.00)	2799 (74.40)	
Yes	64 (32.00)	963 (25.60)	
History of emphysema			0.1726
No	198 (99.00)	3747 (99.63)	
Yes	2 (1.00)	14 (0.37)	
Missing	0	1	
History of chronic bronchitis			0.6110
No	191 (95.98)	3634 (96.65)	
Yes	8 (4.02)	126 (3.35)	
Missing	1	2	
History of asthma ****			<0.0001
No	147 (73.50)	3287 (87.44)	
Yes	53 (26.50)	472 (12.56)	
Missing	0	3	
Close relative had asthma			0.8575
No	155 (78.28)	2921 (78.82)	
Yes	43 (21.72)	785 (21.18)	
Missing	2	56	
Symptoms (≥40 years old)			
Chronic cough			0.3452
No	29 (87.88)	1121 (92.34)	
Yes	4 (12.12)	93 (7.66)	
Missing	167	2548	
Coughing phlegm			0.4748
No	32 (96.97)	1142 (93.99)	
Yes	1 (3.03)	73 (6.01)	
Missing	167	2547	
Wheezing ****			<0.0001
No	159 (79.50)	3386 (90.13)	
Yes	41 (20.50)	371 (9.87)	
Missing	0	5	
Shortness of breath			0.6677
No	24 (72.73)	923 (75.97)	
Yes	9 (27.27)	292 (24.03)	
Missing	167	2547	
Lung function			
FEV_1_, L ****	3.25 ± 0.73	3.56 ± 0.81	<0.0001
FEV_1_, % predicted ****	90.64 ± 11.09	99.32 ± 12.07	<0.0001
FEV_1_ < 80% predicted-no. (%)	34 (17.00)	206 (5.48)	0.0829
FVC, L ****	4.48 ± 1.01	4.34 ± 1.01	<0.0001
FVC, % predicted ****	109.43 ± 17.21	103.07 ± 17.31	<0.0001
FEV_1_/FVC **	0.72 ± 0.02	0.82 ± 0.05	0.0041
Dietary inflammatory index ^1^			
DII ****	3.82 ± 0.26	3.76 ± 0.25	<0.0001

Results are shown as N (%) for binary variables, and as mean ± standard deviation (SD) for continuous variables. ^1^ The DII was calculated per 1000 calories of daily food consumed. * *p* < 0.05, ** *p* < 0.01, *** *p* < 0.001, and **** *p* < 0.0001 for the comparison of individuals with and without Early COPD.

**Table 2 nutrients-14-02841-t002:** Association of the dietary inflammatory index (DII) with Early COPD, NHANES 2007–2012.

Early COPD	β (95% CI) ^1^, *p*-Value			
	Model 1 ^2^ (*n* = 3962)		Model 2 ^3^ (*n* = 3437)	
DII				
Continuous	2.361 (1.335, 4.174) **	0.0031	1.903 (1.034, 3.502) *	0.0386
Quartile 1	1.0 (reference)		1.0 (reference)	
Quartile 2	1.108 (0.712, 1.726)	0.6486	0.960 (0.596, 1.547)	0.8664
Quartile 3	1.436 (0.943, 2.186)	0.0914	1.156 (0.732, 1.823)	0.5345
Quartile 4	1.657 (1.100, 2.496) *	0.0156	1.288 (0.824, 2.015)	0.2667
*p*-trend	1.195 (1.051, 1.359) **	0.0066	1.104 (0.957, 1.273)	0.1757

Notes: In sensitivity analysis, dietary inflammatory index was converted from a continuous variable to a categorical variable (Quartiles). ^1^ β: effect sizes; 95% CI: 95% Confidence interval. ^2^ Model 1: No covariates were adjusted. ^3^ Model 2: Adjusted for age; sex; PIR; health insurance coverage; sedentary activity; smoking exposure (pack-years); mineral dusts; exhaust fumes; wheezing; asthma. *p*-Value: * *p* < 0.05, ** *p* < 0.01.

**Table 3 nutrients-14-02841-t003:** Subgroup analysis.

DII	Sample Size	Early COPD	*p*-Value
Age			
20–29	1153	1.820 (0.735, 4.503)	0.1954
30–39	1187	2.024 (0.699, 5.855)	0.1934
40–49	1097	0.741 (0.146, 3.754)	0.7177
Sex			
Females	1664	2.325 (0.999, 5.410)	0.0502
Males	1773	1.132 (0.435, 2.944)	0.7988
PIR			
<1.85	1576	2.350 (1.010, 5.470) *	0.0474
≥1.85	1861	1.077 (0.416, 2.784)	0.8789
Health insurance coverage			
No	1136	1.878 (0.705, 5.000)	0.2072
Yes	2301	1.690 (0.742, 3.852)	0.2117
Sedentary activity			
<3 h	712	2.259 (0.629, 8.110)	0.2114
3–6 h	1256	1.223 (0.423, 3.542)	0.7100
≥6 h	1469	1.867 (0.678, 5.145)	0.2272
Smoking exposure (pack-years)			
0	2129	1.475 (0.635, 3.429)	0.3664
<10	848	3.689 (1.133, 12.012) *	0.0303
≥10	460	0.417 (0.082, 2.126)	0.2927
Smokers living in the home			
0	1379	0.754 (0.245, 2.317)	0.6219
1–2	551	3.550 (0.946, 13.319)	0.0604
≥3	83	0.129 (0.002, 9.767)	0.3531
Mineral dusts			
No	2293	2.315 (1.117, 4.800) *	0.0240
Yes	1144	0.731 (0.196, 2.721)	0.6401
Fumes from machinery or engines			
No	2529	1.800 (0.890, 3.639)	0.1018
Yes	908	1.416 (0.349, 5.735)	0.6262
Wheezing			
No	2973	1.298 (0.626, 2.693)	0.4833
Yes	464	4.016 (1.157, 13.947) *	0.0286
Asthma			
No	3069	1.702 (0.841, 3.444)	0.1393
Yes	368	1.796 (0.433, 7.452)	0.4197
History of emphysema, bronchitis or asthma during childhood			
No	2549	1.180 (0.556, 2.505)	0.6668
Yes	893	4.277 (1.362, 13.430) *	0.0128

Notes: The results of subgroup analysis. *p*-Value: * *p* < 0.05.

**Table 4 nutrients-14-02841-t004:** The dietary inflammatory index (DII) and lung function parameters, NHANES 2007–2012.

Lung Function Measures	β	95% CI	*p*-Value
All participants	N (missing) = 3440 (522)		
FEV_1_, L	−0.40	(−0.48, −0.32) ****	<0.0001
FEV_1_, % predicted	−4.42	(−6.08, −2.76) ****	<0.0001
FVC, L	−0.55	(−0.65, −0.45) ****	<0.0001
FVC, % predicted	−8.86	(−11.14, −6.58) ****	<0.0001
FEV_1_/FVC	0.01	(0.00, 0.02) **	0.0086
Participants with Early COPD	N (missing) = 180 (20)		
FEV_1_, L	−0.43	(−0.74, −0.12) **	0.0072
FEV_1_, % predicted	−5.22	(−11.46, 1.03)	0.1009
FVC, L	−0.58	(−1.01, −0.16) **	0.0074
FVC, % predicted	−7.83	(−16.77, 1.12)	0.0858
FEV_1_/FVC	−0.003	(−0.011, 0.005)	0.4573
Participants without Early COPD	N (missing) = 3260 (502)		
FEV_1_, L	−0.39	(−0.47, −0.31) ****	<0.0001
FEV_1_, % predicted	−4.15	(−5.85, −2.45) ****	<0.0001
FVC, L	−0.56	(−0.66, −0.45) ****	<0.0001
FVC, % predicted	−9.05	(−11.41, −6.7) ****	<0.0001
FEV_1_/FVC	0.01	(0.01, 0.02) ***	0.0002

Notes: Data presented as β (95% CI), *p*-Value. All models adjusting for age; sex; PIR; health insurance coverage; sedentary activity; smoking exposure (pack-years); mineral dusts; exhaust fumes; wheezing; asthma (in all participants). *p*-Value: ** *p* < 0.01, *** *p* < 0.001, **** *p* < 0.0001.

## Data Availability

Publicly available datasets were analyzed in this study. This data can be found here: https://wwwn.cdc.gov/nchs/nhanes/Default.aspx, accessed on 9 June 2022.

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
