# Peer review of "The Dietary Inflammatory Index and Early COPD: Results from the National Health and Nutrition Examination Survey"

_nutrients, 2022, doi:10.3390/nu14142841_

Round 1

Reviewer 1 Report

I have read the article by Chen et al. with great interest. The authors have investigated the relationship between the Dietary Inflammatory Index and lung function changes in a large cohort of subjects.

Comments:

·       =Was lung function done pre- or post-bronchodilator?

·       =Do you have any data on inhaled medications? Some patients had asthma and ICS is known to reduce lung function decline in COPD.

·       =Do you have any data on the number of chest infection in the last 12 months? If so, was there any association with the DII?

Reviewer 2 Report

The authors found that pro-inflammatory diet contribute to an increased risk of early COPD.

This data is interesting, but it needs valid evidence. The data must certainly be reported in the literature.

I suggest to include in the introduction and in discussion some data of the effects of the microbiota on COPD and on the oxidative stress particularly underlining the part concerning vitamins, vitamin D deficiency:

doi.org/ 10.3390/biomedicines10061337

doi.org/10.1186/s12931-020-01448-3

10.3390/biomedicines10040898

Author Response

Thank you very much for your recognition of the article and your hard work. In response to your valuable suggestions, we will elaborate on the following aspects.

In the third and fourth paragraphs of the Discussion section, from the perspective of the anti-inflammatory (various vitamins and dietary fibers, especially vitamin D) and pro-inflammatory (various lipid components, especially SCFAs) nutrients of DII, the key points are the correlations among nutrients, microbial composition and metabolites, immunity, lung function, and COPD are expounded. Supplementary demonstrations are made based on your suggestions (see the yellow mark).

In addition, regarding the Introduction section, the necessity of "exploring the relationship between early COPD and DII" was discussed. All references are closely related to this study.

Thank you again for your suggestions, all your suggestions are very important, which have important guiding significance for my future research work.